# Urban Sound Auralization and Visualization Framework—Case Study at IHTApark

**Josep Llorca-Bofí** *[ID], **Christian Dreier** [ID], **Jonas Heck** [ID] and **Michael Vorländer** [ID]

Institute for Hearing Technology and Acoustics, RWTH Aachen University, 52062 Aachen, Germany; christian.dreier@akustik.rwth-aachen.de (C.D.); jonas.heck@akustik.rwth-aachen.de (J.H.); mvo@akustik.rwth-aachen.de (M.V.)
* Correspondence: josep.llorca@akustik.rwth-aachen.de

**Abstract:** In the context of acoustic urban planning, the use of noise mappings is a worldwide well-established practice. Therefore, the noise levels in an urban environment are calculated based on models of the sound sources, models of the physical sound propagation effects and the position of the receivers in the area of interest. However, the noise mapping method is limited to sound levels in frequency bands due to missing temporal and spectral information of the sound signals. This, in turn, leads to missing information about the qualitative sound properties, as they can be evaluated in psychoacoustic parameters. Beyond the scope of the classical noise mapping, auralization and physically-based simulation of sound fields can be applied to urban scenarios in the context of urban soundscape analysis. By supporting the auralization technology with a visual counterpart of the urban space, a plausible virtual representation of a real environment can be achieved. The presented framework combines the possibilities of the open-source auralization tool *Virtual Acoustics* with 3D visualization. In order to enable studies with natural human response or for public communication of urban design projects, those virtual scenes can be either reproduced with immersive technologies—such head-mounted displays (HMD)—or using online video platforms and traditional playback devices. The paper presents an overview of what physical principles can already be simulated, which technological considerations need to be taken into account, and how to set up such environment for auralization and visualization of urban scenes. We present the framework by the case study of IHTApark.

**Keywords:** auralization; visualization; soundscape; sustainable urban planning; noise research; virtual reality

## 1. Introduction

Research on the sound in urban environments has a long history. The first studies of community reactions to noise from subsonic aircraft were collected by Bolt, Beranek and Newman in the 1950s. Kryter compiled and compared similar attitudinal and perceptional studies of transportation noise by other researchers in his 1970 book *The Effects of Noise on Man* [1]. Nowadays, environmental noise is objectively assessed using graphical representations such as noise maps while it is furthermore subjectively assessed using listening tests. Noise maps consist of computed values such as the A-weighted equivalent continuous sound level $L_{A,eq}$ [2]. However, noise mapping based on (weighted) sound pressure levels is not sufficient to study noise perception and the effects of noise on society and health: Comparisons of these objective measures with subjective listening test results revealed aircraft noise to be overrated by about 5dB(A) compared to other noise events (rail, road traffic) with the same $L_{A,eq}$ [3].

Since these values are derived—from a technical point of view—from data in frequency domain, the loss of temporal information makes it impossible to analyze the sensation of how they sound for listeners in the urban environment. For this, two branches can

be distinguished: Psychoacoustic parameters and the soundscape approach (Section 2). To see how this discrimination is reflected in the structure of the paper, refer to Figure 1. For deeper understanding of human auditory perception of urban environmental noise, simulations must include spectro-temporal and spatial features that, in turn, enable the listener to experience the sound field as a whole and interact with sounding objects [4]. Finally, since it is known that hearing sensation is influenced by multimodal effects such as audiovisual coupling as impressively described by, e.g., the *McGurk effect* [5], the visual representation of a scene is indispensable for perceptual studies.

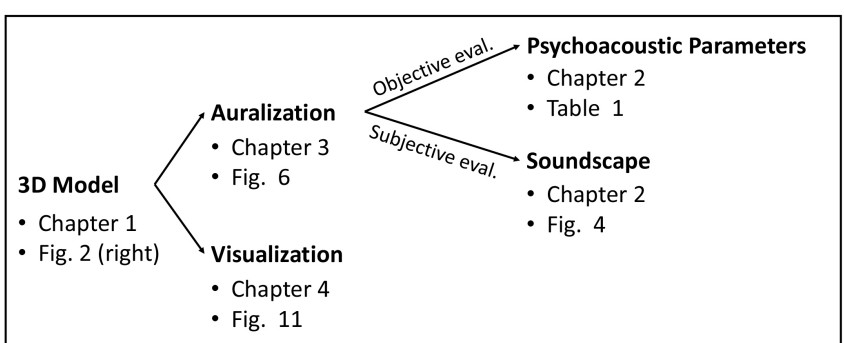

**Figure 1.** Overview of the paper structure according to chapters.

Considering the discussed aspects, this paper presents a combined auralization [6] and visualization framework by the case study at the site referred to as IHTApark (Figure 2). The framework enables controllable conditions for laboratory experiments in sound and noise research while enabling a plausible immersion into the scenery.

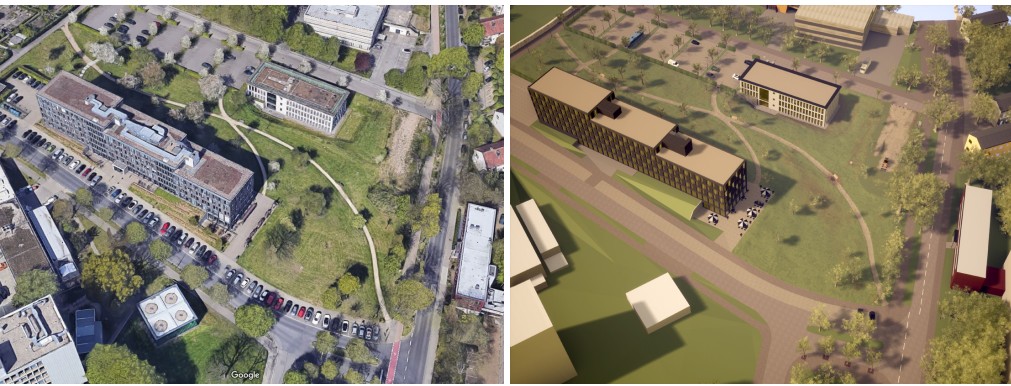

**Figure 2.** IHTApark: Aerial photo of the site (**left**) and 3D visualization export (**right**).

The simulated environment includes fully synthesized technical sound sources—such as cars $S_1$ and aircrafts $S_2$—that are enhanced by recorded natural background sounds, as presented in Figure 3. This scenario is presented for three different weather conditions (summer, rainy and winter), as it is considered by the sound propagation algorithms. Based on this case study, a guideline of the discussed techniques is provided in order to find its application in soundscape and urban sound studies.

The paper is structured as follows: Section 2 introduces the concepts of psychoacoustic evaluation and the soundscape approach. In Section 3, auralization and state-of-the-art outdoor sound propagation simulations are described. Section 4 describes the workflow, elements and techniques to visualize an urban scene in virtual reality. Finally, Section 5 summarizes the results. Section 6 discusses future research and application lines.

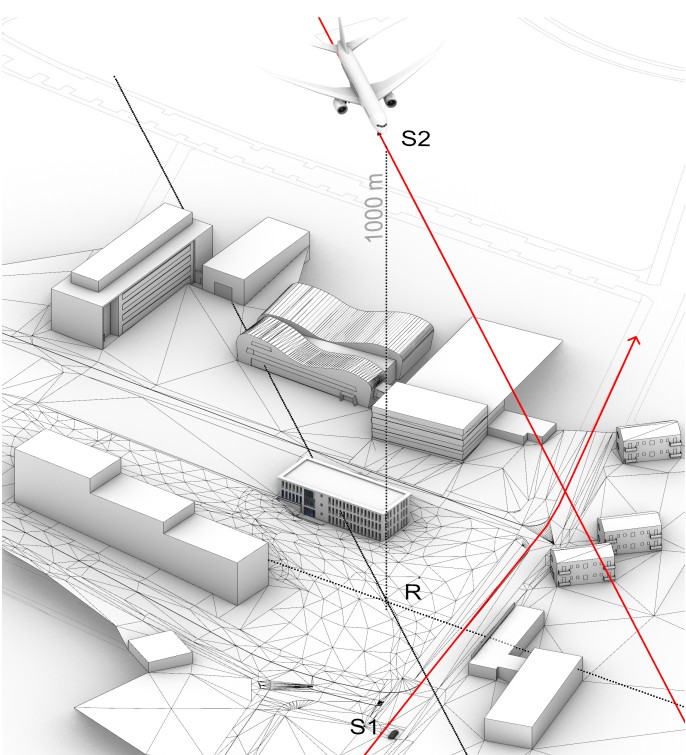

**Figure 3.** IHTApark: Detailed geometry, acoustic sources S1 and S2 and acoustic receiver R. The geographic location of the receiver by coordinates according to ISO 6709 is +50.7807+006.0668.

## 2. Soundscape, Urban Noise Planning and Its Psychoacoustic Evaluation

Soundscape is the approach to extract descriptions of a listener's sound perception in an environmental context (Figure 4). The term was coined by Schafer [7] as a reaction to the emerging field of environmental noise research that manages sound as waste, rather than as a resource. Following The European Landscape Convention [8], soundscape is defined as the acoustic environment of a place, as perceived by people, whose character is the result of the action and interaction of natural and/or human factors. The field of soundscapes overlaps, to various degrees, with fields of acoustics such as sound quality, acoustic comfort in buildings, and music; but also with non-acoustic fields such as wilderness and recreation management, urban and architectural design, as well as landscape planning and management. A review on applications to soundscape planning was published by Brown [9]. A comprehensive book on soundscape in built environment by Kang and Schulte-Fortkamp [10] systematically collects concepts, measurement procedures and different applications of the soundscape approach in planning, design and assessment. Soundscape is standardized by ISO12913 [11].

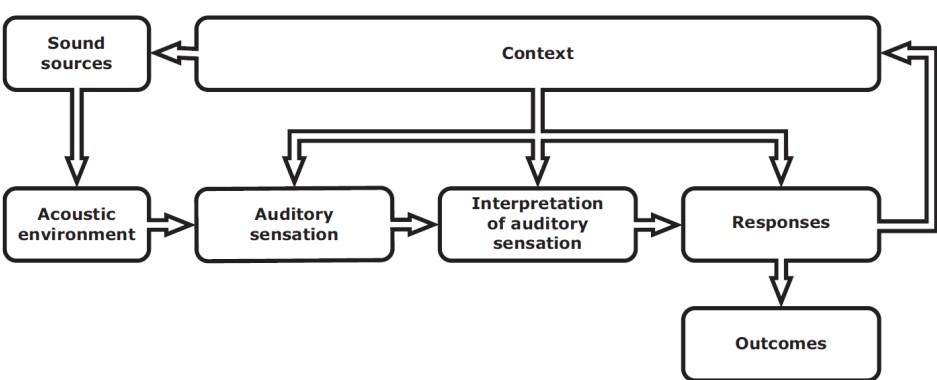

**Figure 4.** Conceptual framework of soundscapes (from ISO 12913-1).

The research field of psychoacoustics aims for quantifiable parameters that describe human hearing sensation. The book by Fastl and Zwicker [12] contains comprehensive research results on that topic. Psychoacoustic evaluation of audios signals can be based on computation of standardized parameters (Table 1). Due to the modular nature of auralizations, the acoustic quality of setups with different design and component modifications can be compared based on calculated psychoacoustic parameters, as exemplarily presented by the authors for partial aircraft noise sources in [13].

**Table 1.** Main psychoacoustic parameters, description and usable standards.

|  | Loudness [Sone] | Sharpness [Acum] | Roughness [Asper] | Fluctuation Strength [Vacil] | Tonality |
|---|---|---|---|---|---|
| Description | Magnitude of auditory sensation | Timbre due to high frequency components | Amplitude modulation between 15 Hz and 300 Hz | Similar to roughness, at slower amplitude modulations | Tonal components characterization |
| Standards | DIN 45631, ANSI S3.4-2007, ISO 532-1 | Aures [14], von Bismarck [15], DIN42692 | ECMA418-2 | Terhardt [16] | DIN 45681 |

From urban and architectural design perspective, the place is the core of all tasks. Different schools of urbanism in the last fifty years constantly recall a definition of urban place as not just a specific space, but comprising all the activities and events which make it possible. Urban theorists such as Gehl [17], Cook [18] and Punter [19] argued that successful urban places would predominantly be based on street life, and the various ways in which activity occurs in and with buildings and spaces, thus configuring different sound environments. In particular, the urban quality of a space might meet the following conditions: development of intensity, mixed use of public space, fine grain, adaptability, human scale, city blocks and permeability, streets contact and visibility, public realm, movement, green spaces, landmarks, architectural style as image. In this regard, urban planners developed ways to present their concepts to people, mainly visual representations grounded on maps. However, representation of activity in such maps is not always easy neither accurate enough to understand the perception of street life. Used by architects and urban planners when presenting their designs, visual renderers are of great help to understand how others would visually perceive their designs. From the acoustic side, auralization techniques can make an acoustic environment audible for a receiver. In this sense, the integration of acoustic representation of urban spaces with activities gives the chance of personal evaluations, beyond studies based on interviews detached from street life and the perception of its complexity by individual perspectives, c.f. [20]. This requires, from the technical side, calibrated models which allows user's evaluation. Using those models, instant comparisons, ratings or preference evaluations can be obtained.

On urban evaluation, there is an increased awareness of assessing not only the physical quantities of public space, but also the societal qualities. The public realm is the social space which comes into being whenever a piece of actual physical space is dominated by social relations among persons, who are strangers to one another [21]. Investigations on the relation between the public realm and soundscape have already provided insights on the impact of sound on the perception of individuals and societies. More recently, researchers derived people's activities from soundscapes, distinguishing the solitary and the socializing individuals in public space [22]. Others used soundscape evaluations to study the effects on restorativeness in urban parks [23]; and described how the soundscape characters of an urban square changed during COVID-19 lockdown [24]. The conclusion is always that soundscapes need to be conceived and investigated by first identifying relevant semantic features that properly describe people's perception before correlating them with quantifiable (acoustic) parameters.

Two elements of the soundscape concept (Figure 4)—sound sources and acoustic environment—are common to the auralization paradigm. On the perceptual side, it supplements the model with the listener's context. Per definition, the context includes the interrelationships between person, activity and place, in space and time. The context may influence soundscape through the auditory sensation, the interpretation of auditory sensation, and the responses to the acoustic environment.

## 3. Auralization of Complex Urban Scenarios

For an auralization of any kind, always three main elements are to be considered: A source, a receiver and propagation paths between them. Of course, multiple sources and multiple receivers can be included. This, in return, increases the computational effort. Each of the elements has its own properties and can be influenced by further effects as shown in Figure 5.

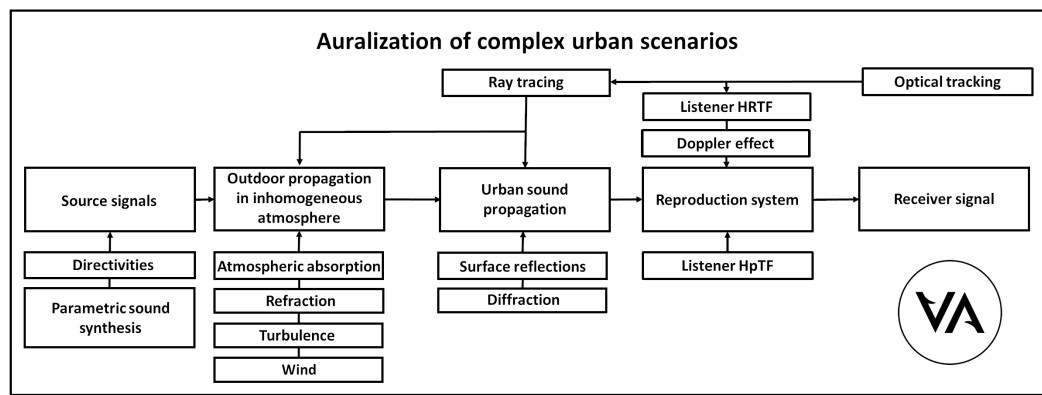

**Figure 5.** Block diagram of environmental noise auralization in VA.

Urban sound propagation can be assumed to be a special case of outdoor sound propagation: In addition to the consideration of physical properties of the propagation medium, i.e., of air, it must be counted for reflections (absorption and scattering) and diffraction. All are material-dependent and vary with respect to frequency and direction of sound incidence. Regarding the model of urban surroundings, the following materials can be clustered:

- ground: asphalt, lawn;
- building façades: concrete, steel, glass.

Usually, from at least one direction (from the sky) no reflection arrives at the receiver, when significant change of weather is neglected. According to the number of reflections and diffractions between the objects of an urban scene, different reflection and diffraction orders are distinguished in ascending order (i.e., 1st order reflection, 3rd order diffraction, etc.). Rendering modules calculate the urban sound propagation according to the following steps: First, during path tracing the propagation paths between source and receiver position must be identified up to a predefined order. Subsequently, each individual path has to be analyzed with respect to its angle of incidence at the receiver and convolved with the corresponding angle-dependent directivity of the receiver. Nevertheless, it is useful to perform so-called perceptual culling, which discards inaudible paths already during path tracing in order to save computing time. For this purpose, the *MyBinauralOutdoorUrban* renderer was implemented in the open-source software *Virtual Acoustics* (www.virtualacoustics.org accessed on 14 January 2022). Since the renderer contains only the propagation-related algorithms, information about the sound sources and receivers as well as the propagation medium must still be added for complete auralization.

We present in the case of IHTApark a hybrid auralization approach. This means that dominant sounds, such as the pass-by noise of individual cars or an aircraft flyover, are synthesized, whereas additional recorded background noise is used to enhance the

perception of real immersion. Details about the elements' properties are discussed more comprehensively in the following.

### 3.1. Transportation Noise Sources

In an urban environment, different technical and natural sounds occur. The latter are discussed in Section 3.4. Technical sounds must be considered in auralizations by individual source models for each source type. We use source models that combine real-time parametrically synthesized signals and corresponding directivities (Figure 5). It should represent the angle-dependent and spectro-temporal characteristics of the sound to be modeled. In literature, source models are found to be based on analytical, numerical or (semi-) empirical models. In this context, the following source modeling approaches should be highlighted as examples:

Cars:

- Engine noise: [25–27];
- Road-tyre noise [28–30].

Aircraft:

- Jet noise [31,32];
- Fan noise [33,34].

Trains:

- Wheel-rail rolling noise: [35–38];
- Rail vehicle powertrain: [39].

The detailed description of our aircraft auralization model that is also used in this framework is published in [40]. Details on the vehicle models will be explicitly presented in upcoming publications.

### 3.2. Outdoor Sound Propagation

As shown in Figure 5, the simulation of sound wave propagation through the atmosphere between the position of transportation noise sources (cars, airplanes, trains) and the listener includes the physical properties of the medium and its geometrical boundaries. Since most ground surfaces—natural materials as well as asphalt—are porous, sound propagation in urban environment is sensitive to the acoustical properties of ground surfaces as well as to meteorological conditions. In case of aircraft noise, stochastic turbulence and shadow zones occur. Notable effects to be considered in urban noise simulation are:

- Spreading loss;
- Atmospheric attenuation;
- Diffraction (at edges and barriers);
- Ground and surface reflections;
- Refraction (wind and temperature gradients);
- Vegetation attenuation;
- Turbulence.

Except for spreading loss, all effects are frequency-dependent. This circumstance, considering the large distances acoustic waves travel, poses special requirements on algorithms for fast auralization of urban scenes: Instead of numerically solving the wave equation, the assumptions of geometrical acoustics (GA) are used. Accepting the trade-off of reduced accuracy in contrast to wave-based modeling, in GA, wave properties of sound are neglected, by assuming the sound waves to propagate as rays. The application of computationally optimized state-of-the art ray tracing algorithms allows atmospheric propagation paths (eigenrays) can be calculated with very low latency [41]. Since diffraction cannot directly be simulated by GA methods, modified diffraction models based on asymptotic formulations—where infinite edge's contribution (the diffraction wave emanating from the edge) are described by an explicit expression—can be used for efficient computation, e.g., the Uniform Theory of Diffraction (UTD) [42]. An open-source tool for finding sound paths

in an urban environment including reflections from building surfaces by computation of a deterministic solution to this problem based on the *image edge model* was developed by Erraji [43].

The processing of multiple sources is computationally costly: For each source, many (~10 to 1000) propagation paths to the receiver have to be calculated. To cope with this complexity, clustering techniques have been developed which aggregate several neighboring sources in order to reduce the number of propagation paths, c.f. [44].

At the example of the IHTApark model (Figure 6), a virtual car is placed at the source position *S* and the receiver is sitting on the bench at position *R*. The visualization in Figure 6 separately indicates the reflected and diffracted sound paths between *S* and *R*. While the reflections occur at the building façades and the ground, the diffractions occur at the building edges.

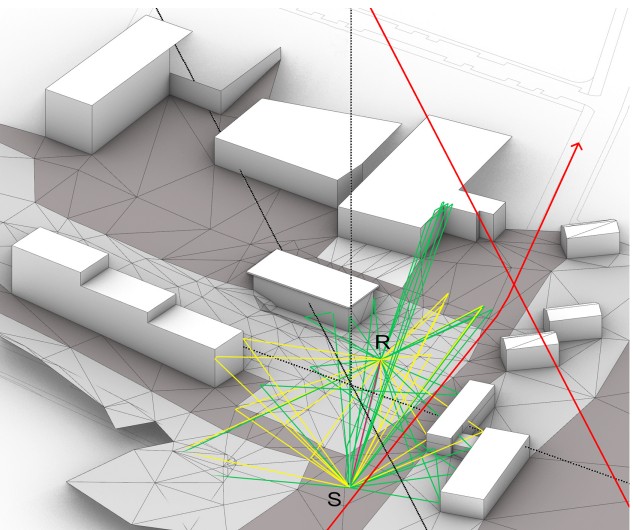

**Figure 6.** Visualization of sound paths in the IHTApark model between a car on the street (S) and a receiver sitting on the bench (R). The result contains second order reflections and diffractions. Yellow marked paths depict stronger diffraction components, green indicates more reflections in the path.

Technically, each sound path contributes an impulse with individual time delay and amplitude to the overall impulse response at the receiver R (Figure 7, left). By application of the Fourier transformation, a transfer function in frequency domain (Figure 7, right) can be derived from the impulse response. It contains a position-dependent amplitude spectrum, which includes the effects of geometrical spreading loss, atmospheric absorption, second order surface reflections and diffraction.

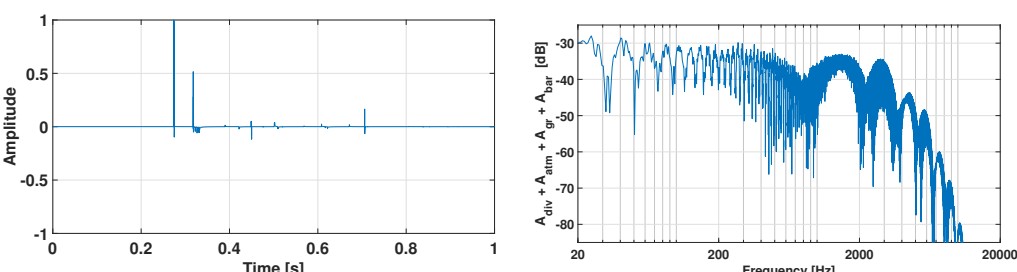

**Figure 7.** Impulse response (**left**, normalized) and according transfer function (**right**) between the positions of source S and receiver R from the scenario in Figure 6.

As mentioned earlier, the transfer functions are dependent on the physical medium parameters humidity, temperature, pressure and wind [45]. The simulation of atmospheric sound propagation considers various frequency-dependent and frequency-independent at-

tenuations. To calculate the total attenuation in decibels, the individual partial contributions are added as follows:

$$A_{\text{tot}} = A_{\text{div}} + A_{\text{atm}} + A_{\text{gr}} + A_{\text{bar}} + A_{\text{misc}}$$

where

$A_{\text{div}}$ = attenuation due to spherical spreading;
$A_{\text{atm}}$ = att. due to atmospheric absorption;
$A_{\text{gr}}$ = att. due to the ground effect;
$A_{\text{bar}}$ = att. due to diffraction;
$A_{\text{misc}}$ = att. due to miscellaneous other effects (e.g., turbulence).

These parameters influence the spectrum of the travelling acoustic wave by frequency-dependent absorption, refraction and advection. The longer the distance between source and receiver, the more significant the change is for sound perception.

The influence of different weather conditions on the noise levels and the psychoacoustic perception of aircraft noise was studied by the authors in [40]. As exemplary scenarios of this paper's case study, three different weather conditions are shown—summer, winter and rainy—to illustrate the influence of the physical medium parameters that are contained in Figures 8 and 9.

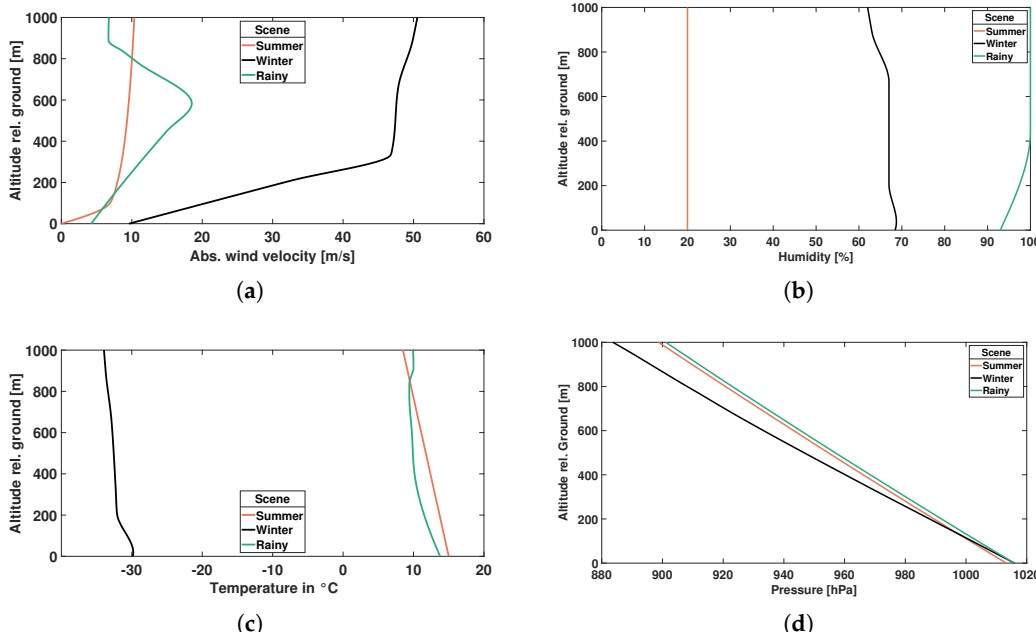

**Figure 8.** Measured (**a**) wind, (**b**) relative humidity, (**c**) temperature, and (**d**) static pressure profiles for simulation of atmospheric sound absorption.

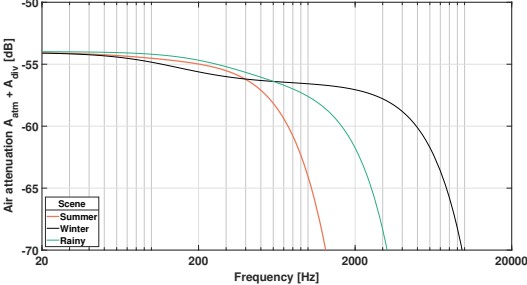

**Figure 9.** Atmospheric transfer functions (including spreading loss, and without ground reflection) for the weather conditions according to the three scenarios (*summer, rainy* and *winter*, see Figure 12). The transfer functions are related to an aircraft position at 1 km altitude straight above the receiver.

### 3.3. Receiver: Virtual Ambisonics Encoding and Playback Decoding

As introduced in Figure 5, the receiver is the last part of an auralization chain. It is crucial to make the simulated sources and propagation paths audible under consideration of the acoustic receiver characteristics. In this last step, it must be decided on the reproduction system (headphone or loudspeaker) and its required accuracy.

Traditionally, in soundscape and environmental noise evaluations, one-channel, omnidirectional microphones have been used in order to determine sound pressure levels and psychoacoustic parameters. When it comes to auditory evaluation of these sounds, the binaural format serves as established standard because it allows for capturing and reproducing sounds like human hearing with two ears. For more details about the concept of binaural hearing, you may refer to Blauert [46] and Litovsky et al. [47]. In recent years though, multi-channel audio formats have further developed: Thanks to higher computation power, they can be used in numerous, even mobile applications. Especially, Ambisonics has become popular as it is easy to be converted into other formats.

The basic idea of Ambisonics comes from the coincidence of multiple microphones with different orientations, c.f. [48]. Such a recording pattern allows to capture direction-dependent information of the recorded signals. This is possible because each signal of a single microphone capsule can be weighted with the capsule's directivity. The four-capsule microphone that is used for background noise recordings, which are introduced in the next section, is shown in Figure 10. The directivity of this microphone was measured following the procedures used by Richter et al. [49].

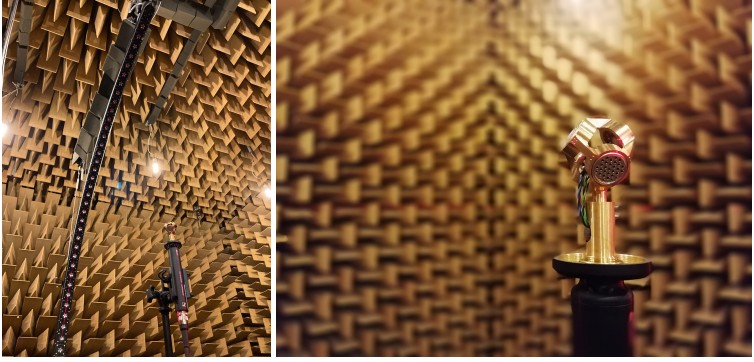

**Figure 10.** Directivity measurement of an Ambisonics microphone in a loudspeaker array (**left**); microphone in close-up view (**right**).

Several formats for Ambisonic encoding exist, most prominently Ambisonics A-format and B-format. The different formats can be transformed into each other. In order to take into account the binaural reproduction system (i.e., headphones), the decoding must consider HRTF (head-related transfer function) high-frequency roll-off for low Ambisonics orders [50,51]. The HpTF equalization is proposed to be done according to [52].

### 3.4. Urban Background Noise

In most publications, traffic and industrial noise are named as predominant and most annoying sound sources in urban environments. Thus, oftentimes they are denoted as urban noise or environmental noise, sometimes as outdoor noise. These terms are generally not well distinguished from each other. For a categorization of environmental sounds, you may refer to Yang and Kang [53] who clustered sounds into the categories of natural sounds and urban sounds. While natural sounds contain sound sources from nature, such as water, wind and birds, urban sounds only contain human-made sounds and sounds emitted from human-built objects. For more details on masking effects as well as psychoacoustic effects of natural sounds, please refer to Yang et al. [54].

With regard to urban background noise in auralizations, the main question is whether to use synthesized sources or insert recorded sounds. In the paragdigm of Figure 5, the

source modelling of natural sound sources is challenging, mainly because measuring their directivities is a complex task (e.g., for birds or random geometry-material interactions in water streams).

Most methods presented in literature use recorded source signals for auralization. Acoustic sources can be created by means of recording and filtering real noise sources, e.g., pass-by noise of cars or wind turbine noise, see Section 3.1. Another approach is to not record the whole source signal but the impulse response only, as Mori et al. [55] did outdoors for loudspeakers used for public address announcements. The number of propagation increases considerably with an increasing number of sources that would need to be modeled for a realistic background noise. This is why here, the additional background sounds are delivered as real recordings in first-order Ambisonics format and added to the rendered scene with synthesized sounds that are also computed in Ambisonics. They are inserted before the final downmix or upmix from Ambisonics to the preferred playback format, usually a binaural signal.

## 4. Visualization Framework: Three-Dimensional Architectural Rendering

The basic paradigm of visual simulation of architectural and urban scenes includes four interconnected prerequisites: geometry, materials, light sources and viewers. The approach to construct the model is highly modular. The modular setup allows for the independent modification of each module under certain variables, independent from the rest of the modules, which makes the model versatile. For example, changes of the sunlight between versions of the same model are created by the modification of the sun position and the depending variation of certain subsequent variables (light color and light intensity, for instance). Modifying the sunlight is independent of the building's geometry, the materials and the receiver. The lighting and shadows applied to the geometry are not backed to it, but calculated for the new configuration. There are two main axes of modularity in the visual model. The first axis corresponds to the four prerequisites of a simulated visual environment. These can be understood when looking to the classical field of computer graphics: geometry, or ways to represent and process surfaces; an imaging or processing acquisition, ways to reproduce the visual—or material—characteristics of objects; rendering, algorithms to reproduce light rays; and animation, ways to represent and manipulate motion, as described by Foley et al. [56].

The features of the visual model have no influence on the acoustic model except that both models have the same dimensions and are referenced to the same point of origin and orientation. The process followed by the authors to produce the urban visualization of the IHTApark in its full features follows a common workflow among architects, as it is described by the authors [57]. In the next paragraphs, a detailed description of the case study is provided, following (Figure 11):

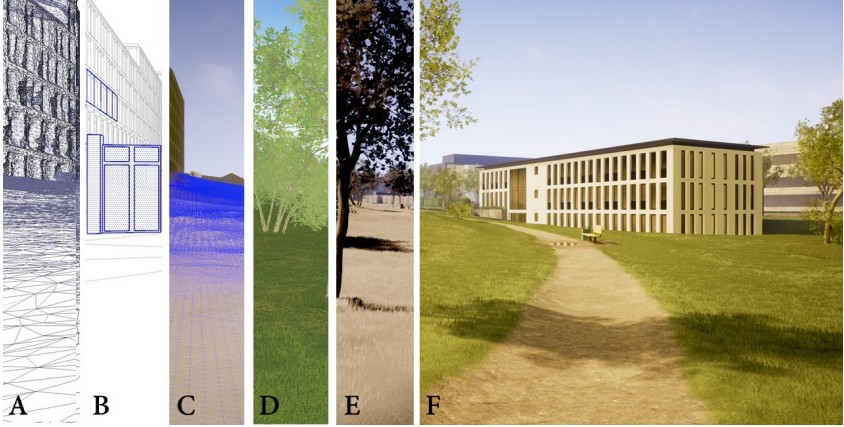

**Figure 11.** Visualization workflow: (**A**) Photogrammetry; (**B**) Modelling and mapping; (**C**) Landscape modelling; (**D**) Materials; (**E**) Lighting; (**F**) Postprocessing.

### 4.1. Photogrammetry

Agisoft Photoscan professional edition and DxO PhotoLab was used in this step. As a first approximation of the real environment, a photogrammetry measurement was done, capturing 200 photos, taken by hand and with the help of a monopod. The mission planning included shooting points and paths, considered two main aspects: the general shape of the surround buildings and terrain should be covered; different shooting heights were desirable. For this reason, the use of the different windows, floors and roofs of both adjacent buildings were used as shooting points. The measurement campaign was conducted in a time range of 3 h at noon, in order to capture similar properties of illumination features. A NIKON D850 camera was used, with fixed parameters: 35 mm focal length of 50 mm, exposure time of 1/60 sec., and F-stop of f/6,3. Two consecutive photographs were overlapping at least with 1/4 of the image. For consecutive shots at different heights, salient visual elements were captured on both images in order to serve as reference. The aperture, velocity and sensitivity of the camera were fixed in order to maintain constant lighting properties for the full model. The photographs were post-processed neutralizing distortion from the objective, using DxO PhotoLab. The processed photographs were then used as input data in the photogrammetry software Agisoft Photoscan, following the next steps: 502/619 images aligned, 3,561,745 points extracted in a cloud, 1,121,284 faces building a mesh. A FBX mapped mesh file was exported from Agisoft Photoscan as final photogrammetry and used later to define the terrain details, especially altitude and elevation. It is is freely accessible via *Zenodo* (https://doi.org/10.5281/zenodo.4629759, accessed on 14 January 2022).

### 4.2. Modelling

Trimble SketchUp 2020 was used in this step. The light propagation implies two additional elements to be computed: the objects' geometry and the visual properties of the materials. In contrast to the previous step, the model was constructed of object-based geometries, meaning that each polygon is defining a specific object; from very small geometrical details, such as fence bars, to large surfaces defining the roof of a building. In order to maintain the hierarchy of objects, the geometries were drawn in SketchUp and grouped, defined by hierarchical objects. In this sense, a building is a group object, containing another group of objects called windows; each window group contains groups of prisms representing the window frameworks, etc. This step can be seen in Figure 11, detail B. The level of detail of this model depends on the distance from the objects to the receiver. Since the receiver will be always placed on the green area between the two big buildings, the façades of both buildings were modelled with a level of detail of 1–2 cm. The terrain was modelled as a mesh following the topography obtained from the photogrammetry mesh (see Section 4.1). In the far field, the sky was modelled as a sphere of 1000 m of diameter, centered at a fixed position close to the user. The urban furniture, trees and leaves were not modelled in SketchUp, but later inserted in Unreal Engine (see Section 4.4). When the modelling task was completed, the model was exported in FBX format and saved as separate file.

### 4.3. Mapping

Autodesk 3ds Max was used in this step. The FBX file maintains the grouping hierarchy introduced in SketchUp, which allows for its recognition in 3ds Max. It can be imported directly in the software mantaining the origin point (0,0,0), orientation and scale. As a checking indicator, each single group should be visible as separate object in the objects list panel. After selecting the whole model (or pressing key Q while dragging left click over the geometry), a *UVW Map* modifier was added to everything that was selected. This modifier applies a texture mapping geometry over all the selected geometries. Choosing a box of 2 m of width and height as the mapping projection assures a constant scaling of the textures mapped on all the geometries. If geometries different to aligned and parallel prisms would be present, a different mapping projection could help. Since the photographs and downloaded textures that are later used for texturing are not necessary scaled to a

real-world map size, we rejected this option. Afterwards, changing the pivot position of each object to (0,0,0) allowed us to make modifications into the 3ds Max Model and export them into Unreal Engine recursively in the exact position. In order to change all the pivots points to (0,0,0), select all the geometry (Ctrl+A), search for *Pivot selection* in the working tab, and select *Affect Pivot Only*. Then, write down the new coordinates to (0,0,0) in the bottom bar line. Finally, *Collapse* command, under the *Utilities* panel, allows the software to manage all objects in a single entity. With the *Modifier Stack Result* option selected, click on *Collapse selected*. The same method applies to the *Reset Xform* option. At this point, by selecting everything, the model can be exported to FBX format. This model was imported into Unreal Engine 4.26, using the option *Import into level*. Importing options were *Create Level Actors* and *Importing Normals*.

### 4.4. Landscape, Trees and Foliage

Unreal Engine 4.26 was used in this step. The landscape was redefined using the landscape sculpture tools in the software to finally define the details of paths, small hills and curvatures. On top of the landscape, the materials of grass and ground were added as procedural materials. Further details such as puddles, steps and bike marks were added as independent objects. The trees and foliage were added and painted on top of the landscape using the assets from the *City Park Environment Collection* [58] and the *Winter Forest Collection* [59].

### 4.5. Materials

Unreal Engine 4.26 and Adobe Photoshop were used in this step. Two types of materials were used in the model: procedural and physically-based (PBR) ones. Generally speaking, building materials were defined as PBR materials, with photographs as albedo definitions, whereas foliage and trees were defined as procedural materials. The photographs and textures were freely downloaded from *textures3D* [60] and mostly postprocessed and modified in Adobe Photoshop. The scale of the photographs corresponded to the UVW Map of $2 \times 2 \times 2$ m defined in Section 4.3. An additional texture scaling function was added to specific sized textures. Reflective materials additionally include a gray-scale diffusion map.

### 4.6. Lighting

Unreal Engine 4.26 was used in this step. The light is basically what makes objects visible to our eyes. From the physical point of view, it can be described by electromagnetic waves in the visible spectrum. In outdoor scenarios there is a predominance of sun light, due to its higher intensity relative to artificial illuminations. However, considering the sun as the only light source in urban environments is not sufficient to plausibly represent the visual aspects of it. This is why the sky light is also taken into consideration. The sky luminance distribution over the hemisphere depends on the weather conditions, cloud density and position of the sun. The sky light distribution of the clear and overcast distributions defined by the *CIE Standard Overcast Sky and Clear Sky* are considered [61]. The sun light and skylight cast direct illumination as well as indirect illumination, thus generating shadows in the model. Other light sources are included for particular effects, such as artificial omnidirectional point lights for the cloudy versions of the IHTApark in some rooms of buildings, and the airplane warning lights. Dynamic illuminations were used since the movement of shadows is an important factor in outdoor environments. Direct illumination from a *Directional light* as the sun, was linked to the *Sky*, in which *Colors depend on the sun*, thus manipulating the light temperature of the sky according to the sun position. The *Sky* also allows for cloud density and velocity definitions.

Considering the large distances light travels, poses special requirements on algorithms for fast visualization of urban scenes: instead of calculating the representation of the light propagation with all the possible paths, the assumptions of real-time rendering are used. Accepting the trade-off of reduced accuracy in contrast to a full path-tracing modelling, the

real-time rendering merges some optimized approaches. Direct and indirect lighting can be pre-computed for the not-movable geometries using lightmaps [62]. Since movable objects receive dynamic illuminations, those objects require more computational power and are reduced to the minimum number of surfaces. A visualization of the sky and computation of its illumination on the global scene is implemented using image-based lighting (IBL) [63,64]. In order to compute in real time the reflections of light on highly reflective surfaces, such as glass, windows and water, image-based reflections are used. A global illumination (GI) [65] is added to illuminate the hidden surfaces to the indirect illumination. Since the exposure of each point in a scene to the ambient lighting cannot be simulated by the direct and indirect illumination, ambient occlusion (AO) models based on the accessibility value each surface point [66].

The simulation of light propagation through the atmosphere—*light transport* in computer graphics—includes geometrical and physical properties of the medium and its boundaries. Since most ground and building surfaces are diffuse, light propagation in the urban environment is sensitive to the reflection and diffraction properties of those surfaces as well as to meteorological conditions. Notable effects to be considered in urban light simulation are:

- Spreading loss for artificial lights;
- Solar irradiance;
- Light temperature;
- Ground and surface reflections;
- Atmospheric scattering;
- Luminance distribution over the sky hemisphere;
- Fog attenuations and other effects.

After all the geometry was imported, a check on the lightmap resolution for each object was done. The *Lightmap Density* colors indicate whether there are objects which need an increase of such resolution. This is important when the lighting and shadows details are relevant in such an object. Specific object's lightmap resolution can be changed in the object's properties.

### 4.7. Postprocessing and Export of Visual Data

Unreal Engine 4.26 was used in this step. Once the model was finished, postprocessing was used to produce the different weather conditions (Figure 12). For the summer weather condition, *Saturation*, *Light Tempterature* was decreased towards 2700 Kelvin and the relation between *Intensity of Direct Lighting* and *Intensity of Skylight Lighting* was increased. For the rainy weather condition, *Saturation* was decreased, *Light Tempterature* was increased towards 5000 Kelvin grades and the relation between *Intensity of Direct Lighting* and *Intensity of Skylight Lighting* was decreased. For the winter weather condition, *Saturation*, *Light Tempterature* was decreased towards 2700 Kelvin and the relation between *Intensity of Direct Lighting* and *Intensity of Skylight Lighting* was increased; additionally, a snow material was added to the floor and objects, a particle generator was added for the snow drops and trees were updated by the *Winter Forest* package in Unreal Engine.

The visual models were exported as 360 videos using the *Panoramic Capture* plugin [67], at 30 frames/second and a resolution of 4096 × 2048 pixels (4K) resolution. The virtual viewers are geometrical projections of the 3D data, which differ in a number of aspects to what a real eye perceives in the world; the adaptation, anticipation and interpretation capacities are relatively diminished. For this, the geometric properties of a virtual camera are defined: the position in the scene, the orientation, the aspect ratio and the view angle. In the case of the IHTApark environment, the camera can be freely placed at any place. For the video demonstrator presented in Section 5, a trajectory following the central path has been used; the direction of the camera is looking constantly to the South-West, the aspect ratio is defined as 360° images.

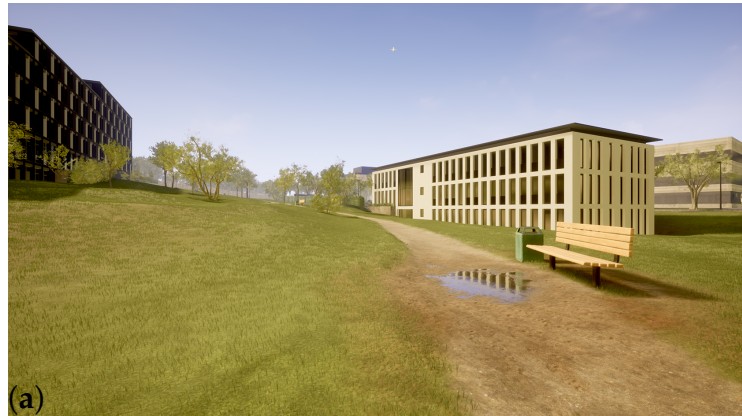

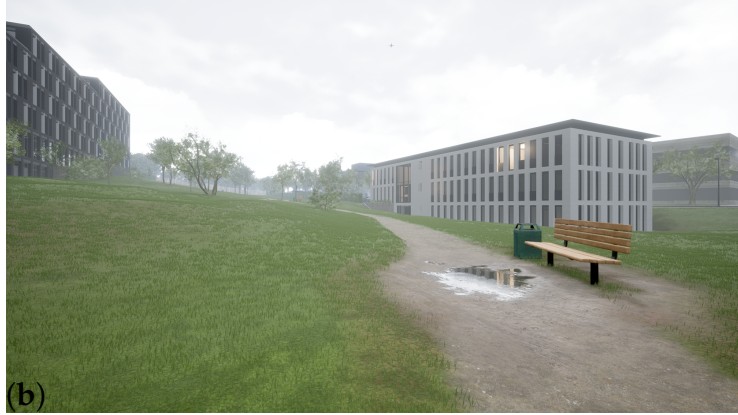

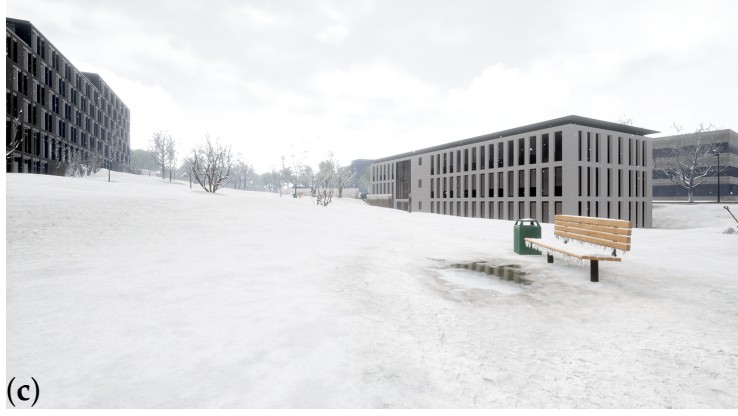

**Figure 12.** Visualization of the IHTApark under three different weather conditions from the same point of view: (**a**) summer, (**b**) rainy and (**c**) winter.

*4.8. Export of Audio-Visual Data*

Since the 3D model is prepared in Unreal Engine, the audio-visual data can be presented over different methods, for example:

- Youtube 360 video player. This method is used as final result in this paper and explained in detail below.
- Head-Mounted Displays (HMD) with headphones. This method was used in [68], following the methods used by Nykänen et al. [69].

Adobe Premiere CC 2020 was used in the following steps. The Ambisonics audio file and the 360 degree video were imported into Adobe Premiere, to be postprocessed and finally uploaded as a 360° video Youtube. For this, a new sequence for VR was created in Premiere, as monoscopic and 1920 × 960 pixels resolution. The resolution sequence was modified adapting it to the raw material, and the audio sampling rate was set to 48,000 Hz.

The number of channels per track was changed to *Adaptive* in the properties of the sequence. With the Audio Mixer window open, the Ambisonics .wav file was imported to a new track. A Master Track was created and set to be able to read 4 channels. This sequence was ready to be exported once the audio and video materials were placed and edited. The exported video was finally postprocessed through the *Spatial Media Metadata Injector* [70] provided by Youtube before the upload.

## 5. Results

The auralization and visualization of the environment is available as a 360 degree Youtube video: https://youtube.com/playlist?list=PLjVMT5BkCe83h299e3FWNivk-MFtRnpw- (accessed on 14 January 2022).

## 6. Discussion

The presented results open the topic to further research and application lines. In particular, it enables environmental noise research by means of spatial auralization and visualization techniques. These techniques can be used to enhance urban planning by plausible sound simulations. Furthermore, from the practitioner's point of view, the availability of a comprehensive workflow from raw data to a Youtube video, shows an easy accessible dissemination of auralization and visualization results in online formats. However, future work on migration of the framework to interactive browser-based formats would increase the degrees of freedom for the listeners.

With regard to the playback of the rendered audio signals, headphone reproduction is the most easy accessible technology. Formatting from Ambisonics formats to binaural headphone format, however, is a standard procedure in spatial audio processing. This also enables to extend the intermediate 3D audio processing from first-order Ambisonics to higher-order Ambisonics (HOA), thus improving the spatial resolution of the acoustic representation.

An additional feature of this work is the possibility to extend the presented auralization methods and sound source models to other methods (such as finite element method) by substituting some modules of the auralization framework (Figure 5). The substitution methods must fulfil the input and output requirements of the framework to be correctly linked to the rest of the modules.

Despite the mentioned advantages, some limitations have to be considered: Even though we introduced calibrated models and recordings, as well as physically-based sound source and sound propagation models, the playback quality at the final user can significantly suffer due to e.g., non-equalized reproduction systems, missing (head)tracking systems or any device of poor quality within the chain. Furthermore, the context is not only important when it comes to the judgement of people regarding the place under investigation, but also regarding the soundscape they expect, which might vary depending on cultural backgrounds.

## 7. Conclusions

A combined framework of physically-based auralization and visualization of urban scenarios was presented in a case study at the IHTApark model. The framework incorporates both recorded and simulated sounds as input data, bridging the gap between both established approaches to urban sound: environmental noise and soundscape. Additionally, the framework is highly modular, enabling the work in packages and compatible formats. Finally, the results can be presented via accessible channels such as Youtube, thus facilitating the content to be made easily available to non-expert users.

**Author Contributions:** Conceptualization, J.L.-B. and C.D.; methodology, J.L.-B., C.D. and J.H.; data curation, J.L.-B., C.D. and J.H.; writing and editing J.L.-B., C.D. and J.H.; supervision and reviewing M.V.; funding acquisition, J.L.-B. and M.V. All authors have read and agreed to the published version of the manuscript.

**Funding:** This research was funded by the Federal Ministry of Education and Research (BMBF), the Federal Ministry for Economic Affairs and Climate Action (BMWi) and the Ministry of Culture and Science of the German State of North Rhine-Westphalia (MKW) under the Excellence Strategy of the Federal Government and the Länder.

**Institutional Review Board Statement:** Not applicable.

**Informed Consent Statement:** Not applicable.

**Data Availability Statement:** IHTApark. Multi-detailed 3D architectural model for sound perception research in Virtual Reality (Version 2). Available at: https://doi.org/10.5281/zenodo.4629759 (accessed on 14 January 2022).

**Acknowledgments:** The authors acknowledge the support of Jonas Kempin for his valuable help in creating the 3D models.

**Conflicts of Interest:** The authors declare no conflict of interest. The funders had no role in the design of the study; in the collection, analyses, or interpretation of data; in the writing of the manuscript, or in the decision to publish the results.

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
