# Peer review of "Urban Sound Auralization and Visualization Framework—Case Study at IHTApark"

_sustainability, doi:10.3390/su14042026_

Round 1

Reviewer 1 Report

The manuscript by Llorca-Bofí et al reports a comprehensive framework that can represent a physical scene with both sound (auralization) and light (visualization). The paper is well organized and the details on constructing each element in the model are presented clearly. The attached video accurately reflects the acoustic effects modeled by the proposed framework, validating its effectiveness. Overall, I think this work is a good contribution to the field and I enjoyed reading it. The manuscript is already in a very good form, and I only have a few comments.

  1. The authors used ray tracing model for the aurolization. While this approach is valid for the study at IHTApart, is the framework work compatible with other methods (e.g., finite element method) for a smaller space and better accuracy? Also, I wonder how much time does it take to complete the aurolization process. Some discussions on this could be helpful as real-time rendering of the acoustic fields is always desirable in games, VR, etc.
  2. In Sec. 4.5, the materials used in the model were divided into two categories, the procedural and physically-based ones. Do procedural ones have any effect on the acoustic wave propagation at all in the model?
  3. In Fig.3, the car is annotated as St rather than S1.
  4. A few typos: Line 318, is it between 11am and 3pm? Line 330, there are two “is” in the sentence.

Author Response

REVIEWER 1: The manuscript by Llorca-Bofí et al reports a comprehensive framework that can represent a physical scene with both sound (auralization) and light (visualization). The paper is well organized and the details on constructing each element in the model are presented clearly. The attached video accurately reflects the acoustic effects modeled by the proposed framework, validating its effectiveness. Overall, I think this work is a good contribution to the field and I enjoyed reading it. The manuscript is already in a very good form, and I only have a few comments.

AUTHOR'S RESPONSE: the authors would like to thank the reviewer for the interest and helpful comments and would like to address the specific comments in the following revision.

  • 1. REVIEWER: The authors used ray tracing model for the auralization. While this approach is valid for the study at IHTApark, is the framework work compatible with other methods (e.g., finite element method) for a smaller space and better accuracy? Also, I wonder how much time does it take to complete the auralization process. Some discussions on this could be helpful as real-time rendering of the acoustic fields is always desirable in games, VR, etc.

AUTHOR'S RESPONSE:

The authors would like to clarify that the used geometrical acoustics (GA) methods are specifically designed for bigger spaces / high frequencies with good accuracy. The framework is modular and can be enhanced by other methods (such as FEM or FDTD results) by substituting some modules of the auralization framework (figure 5). The substitute methods must deliver the input and output formats of the framework to be correctly linked to the rest of the modules. Generally, the GA method itself is optimized towards real-time capability, whereas the modelling process of course has to be done separately beforehand and takes much more time, depending on model accuracy and parameter number. This comment has been added to the discussion (section 6).

The applications and use of such techniques are already included in references 43 and 44, 57, and 69.

  • 2. REVIEWER: In Sec. 4.5, the materials used in the model were divided into two categories, the procedural and physically-based ones. Do procedural ones have any effect on the acoustic wave propagation at all in the model?

AUTHOR'S RESPONSE: The visual materials are divided into two categories in the paper, which has no impact on the acoustic wave propagation at all in the acoustic model. The features of the visual model have no influence on the acoustic model except that both models have the same dimensions and are referenced to the same point of origin and orientation.

  • 3. REVIEWER: In Fig.3, the car is annotated as St rather than S1.

AUTHOR'S RESPONSE: The figure is corrected and updated.

  • 4. REVIEWER: A few typos: Line 318, is it between 11am and 3pm? Line 330, there are two “is” in the sentence.

AUTHOR'S RESPONSE: The typos are corrected.

Reviewer 2 Report

This paper presents a framework to create an urban sound environment with auralization and visualization techniques. The reviewer enjoyed Youtube video showing a created sound environment at IHTApark with the present framework. It was very interesting, but the authors did not provide detailed information to judge the plausibility of the created scene. The descriptions in the auralization part are too simple to publish this article as a scientific paper. Therefore, this reviewer would like to suggest improving the quality of the manuscript with the following suggestions.

1. Figure 7: The authors say that the results include the effect of atmospheric absorption. Please explain how it is included within the manuscript.
2.  Figure 8: What do A_atm and A_div stand for in the vertical axes? Also, how are atmospheric transfer functions calculated? Please explain the calculation procedure in detail. Also, it should be explained what difference is among Summer, Wet, and Snow in physical parameters.
3. The Youtube video results are very interesting, but the reviewer wants to know how the authors examine the plausibility of created sound environments. Please explain it within the result section so that readers can understand its plausibility. Is it based on how readers feel the created scene?
4. Overall, for the auralization part, the authors did not present technical details to the given case. However, its detail is essential for readers to understand the plausibility of the created scene provided by the Youtube video. How is the noise source of cars modeled? How are moving sound sources modeled? How are ground surfaces, including vegetation attenuation, modeled? How is atmospheric attenuation modeled? Please explain their details within the manuscript for readers' deep understanding.
5. Please make a conclusion section and describe the novelty of present work against previous works.

Author Response

REVIEWER: This paper presents a framework to create an urban sound environment with auralization and visualization techniques. The reviewer enjoyed Youtube video showing a created sound environment at IHTApark with the present framework. It was very interesting, but the authors did not provide detailed information to judge the plausibility of the created scene. The descriptions in the auralization part are too simple to publish this article as a scientific paper. Therefore, this reviewer would like to suggest improving the quality of the manuscript with the following suggestions.

AUTHOR'S RESPONSE: the authors would like to thank to the reviewer for the interest and helpful comments and would like to address the specific comments in the following revisions. The authors would like to add that some of the reviewer’s comments are addressed in the references included in the text, which are also specified below.

  • 1. REVIEWER: Figure 7: The authors say that the results include the effect of atmospheric absorption. Please explain how it is included within the manuscript.

AUTHOR'S RESPONSE: Comprehensive analysis of atmospheric absorption effects are presented in the referenced journal papers by some of the authors:  41 and 43. Additionally, an explanation of the atmospheric sound propagation calculation has been included in section 3.2, and a new figure (Figure 8) has been added, showing the atmospheric values used in the study.

  • 2. REVIEWER: Figure 8: What do A_atm and A_div stand for in the vertical axes? Also, how are atmospheric transfer functions calculated? Please explain the calculation procedure in detail. Also, it should be explained what difference is among Summer, Wet, and Snow in physical parameters.

AUTHOR'S RESPONSE: The changes introduced in the previous comment cover these questions. Note: The terms “summer, wet and snow” were replaced throughout the manuscript by “summer, rainy and winter”.

  • 3. REVIEWER: The Youtube video results are very interesting, but the reviewer wants to know how the authors examine the plausibility of created sound environments. Please explain it within the result section so that readers can understand its plausibility. Is it based on how readers feel the created scene?

AUTHOR'S RESPONSE: The effects on the perceived sound quality (roughness and sharpness) were already investigated by the authors, as it can be found in reference 40.

  • 4. REVIEWER: Overall, for the auralization part, the authors did not present technical details to the given case. However, its detail is essential for readers to understand the plausibility of the created scene provided by the Youtube video. How is the noise source of cars modeled? How are moving sound sources modeled? How are ground surfaces, including vegetation attenuation, modeled? How is atmospheric attenuation modeled? Please explain their details within the manuscript for readers' deep understanding.

AUTHOR'S RESPONSE: Detailed model descriptions on the used aircraft models, ground and atmosphere models are given in the above-mentioned journal publications by the author’s. Details on the vehicle models will be explicitly presented in upcoming publications.

  • 5. REVIEWER: Please make a conclusion section and describe the novelty of present work against previous works.

AUTHOR'S RESPONSE: Short conclusion has been included in the manuscript.

Round 2

Reviewer 2 Report

Many thanks for your revision. I still have an important comment about your reply.

A1) AUTHOR'S RESPONSE: Detailed model descriptions on the used aircraft models, ground and atmosphere models are given in the above-mentioned journal publications by the author’s. Details on the vehicle models will be explicitly presented in upcoming publications.
Q1) I couldn't find the explanation about "Details on the vehicle models will be explicitly presented in upcoming publications." within your revised version. That is a critical point for reviewers and readers. Please explain it explicitly in the article. 

Author Response

Dear reviewer,

Thank you for the comments. We included the sentence "Details on the vehicle models will be explicitly presented in upcoming publications." on section 3.1.

The authors